# Bioinformatics analysis of genes associated with disulfidptosis in spinal cord injury

**Shuang Wang**◉[☉], **Xinhua Liu**[☉], **Jun Tian**◉[☉]*, **Sizhu Liu**[☉], **Lianwei Ke**[☉], **Shuling Zhang**[☉], **Hongying He**[☉], **Chaojiang Shang**[☉], **Jichun Yang**[☉]

Shangnan County Hospital, Shangnan County, Shangluo City, Shaanxi Province, China

☉ These authors contributed equally to this work.
* tianjun19730607@163.com

**Data Availability Statement:** The datasets analyzed during the present study are available in the GEO database (http://www.ncbi.nlm.nih.gov/geo/).

## Abstract

Research findings indicate that programmed cell death (PCD) plays a pivotal role in the pathophysiology of spinal cord injury (SCI), and a recently discovered form of cell death, disulfidptosis, has emerged as a novel phenomenon. However, the characterization of disulfidptosis-related genes in SCI remains insufficiently explored. We retrieved SCI-related data from the Gene Expression Omnibus (GEO) database and identified three key genes associated with disulfidptosis in human SCI (*CAPZB*, *SLC3A2*, and *TLN1*), whose mediated signaling pathways are closely intertwined with SCI. Subsequent functional enrichment analysis suggested that these genes may regulate multiple pathways and exert corresponding roles in SCI pathology. Moreover, we predicted potential targeted drugs for the key genes along with their transcription factors and constructed an intricate regulatory network. CIBERSORT analysis revealed that *CAPZB*, *SLC3A2*, and *TLN1* might be implicated in modulating changes within the immune microenvironment of individuals with SCI. Our study provides compelling evidence confirming the significant involvement of disulfidptosis following SCI while offering valuable insights into its underlying pathological mechanisms.

## 1. Introduction

The spinal cord plays a crucial role in transmitting nerve signals to regulate both motor and sensory functions. However, when the spinal cord sustains damage, it can result in irreversible impairment of motor, sensory, and sphincter functions [1, 2]. Severe spinal cord injuries (SCIs) lead to lifelong disabilities that necessitate extensive treatment and long-term care expenses. Consequently, this has profound implications for patients, their families, and society as a whole [3].

In general, the pathophysiology of SCI typically encompasses primary and secondary injuries [4, 5]. The primary injury leads to hemorrhage, ischemia, edema, hypoxia, as well as neuronal and glial cell death. Secondary injury involves intricate pathophysiological mechanisms such as ion imbalance, oxidative stress caused by free radicals, inflammatory responses, and formation of glial scars [6]. Currently, there are no effective clinical treatments available for SCI [7, 8]. The focus of SCI treatment lies in neuroprotection and neural regeneration [9], aiming to minimize neuronal and glial cell death after SCI.

**Funding:** The author(s) received no specific funding for this work.

**Competing interests:** The authors have declared that no competing interests exist.

In recent years, programmed cell death (PCD) has emerged as a pivotal process subsequent to SCI, wherein the early onset of PCD in acute SCI exacerbates the deterioration of spinal cord function [10]. Recent advancements have unveiled diverse forms of PCD, encompassing apoptosis, necroptosis, autophagy, ferroptosis, pyroptosis, and paraptosis, all intricately linked with neuronal and glial cell demise following acute SCI [11–16]. Furthermore, extensive investigations have elucidated the underlying molecular mechanisms governing these distinct modes of cell death in SCI.

Recently, a novel form of cell death called disulfidptosis has been proposed in research [17]. Unlike existing forms of PCD, disulfidptosis is triggered by the accumulation of excess cysteine in cells, leading to disulfide stress. Cells with elevated levels of *SLC7A11* can uptake cysteine through mechanisms mediated by *SLC7A11*, thereby preventing ferroptosis under conditions of glucose deprivation. However, this process may result in a distinct form of cell death known as disulfidptosis [17, 18]. This rapid mode of cell demise typically occurs when cells experience glucose deprivation [19, 20]. Neurons and glial cells within the ischemic penumbra after spinal cord injury are subjected to ischemia-hypoxia. Whether disulfidptosis is involved in the pathophysiological processes following SCI has not been thoroughly investigated.

To gain a comprehensive understanding of the gene regulatory mechanisms associated with disulfidptosis in SCI patients, we employed bioinformatics technology and experimental verification to identify three genes (*CAPZB, SLC3A2,* and *TLN1*) that are implicated in disulfidptosis after SCI. Further analysis revealed that these three genes regulate the pathophysiological processes following SCI through multiple pathways. Additionally, our findings suggest their potential involvement in various immune responses post-SCI. Our research findings offer valuable insights for further investigation into disulfidptosis in SCI.

## 2. Materials and methods

### 2.1 Data sources

The datasets GSE45006, GSE104317, GSE166009, GSE182796, GSE183591 and GSE234774 were obtained from the GEO database (https://www.ncbi.nlm.nih.gov/geo/). Inclusion criteria for data selection included samples collected within 7 days of SCI. The GSE45006 dataset (GPL1355) comprises *rat* spinal cord tissue RNA microarray data from 4 control samples and 12 SCI samples. The GSE104317 dataset (GPL1355) includes *rat* spinal cord tissue RNA microarray data from 4 control samples and 4 SCI samples. The GSE166009 dataset (GPL27597) contains *rat* spinal cord tissue RNA microarray data from 3 control samples and 9 SCI samples. Similarly, the GSE182796 dataset (GPL23040) consists of *rat* spinal cord tissue RNA microarray data from 5 control samples and 5 SCI samples. Additionally, the GSE183591 dataset (GPL4135) provides *rat* spinal cord tissue RNA microarray data with a total of 4 control samples and 4 SCI samples. For spatial transcriptomic analysis in *mouse* SCI, we utilized the spatial transcriptomic dataset named as GSE234774-GSM7474525. Among these datasets, GSE45006, GSE104317, andGSE166009 were considered as the experimental analysis cohort comprising a total of 11 control group samples and 25 SCI samples. GSE182796 and GSE183591were used as a validation cohort with a total of 9 control group samples and 9 SCI samples. Further validation of DRGs in SCI was performed using spatial transcriptomic data from the GSE234774-GSM7474525 dataset. 24 DRGs (*ACTN4, CAPZB, CD2AP, DSTN, FLNA, FLNB, INF2, IQGAP1, MYH10, MYH9, MYL6, PDLIM1, SLC7A11, TLN1, ACTB, SLC3A2, RPN1, OSXM, NUBPL, NDUFS1, NDUFSA11, NCKAP1, LRPPRC, GYS1*) [17, 21] were selected from previous articles for further analysis.

## 2.2 Data consolidation and differential analysis

Data processing was performed using the "limma, version 3.58.0 " and "sva, version 3.56.0 " packages in R language (version 4.4.0). The experimental and validation data were merged, batch effects were eliminated, and differential analysis was conducted [22, 23]. To identify DEGs between subgroups (SCI and normal), the "limma" package in R software was employed for differential analysis of the experimental and validation datasets [23], Limma is based on linear models and uses weighted least squares to estimate differences in gene expression, and corrects for multiple testing issues using Bayesian methods. Visualization was carried out using the "ggplot2, version 3.5.1," "pheatmap, version 1.0.12," "ggrepel, version 0.9.6," and "dplyr, version 1.1.4," packages [24]. Considering the potential significant impact of minor changes in the nervous system, a threshold of adj.P $\leq$ 0.05 was set for selecting differentially expressed genes to ensure comprehensive analysis [25]. All R packages used in the article were set to their default parameters.

## 2.3 GO and KEGG analysis of SCI-DEGs

The SCI-DEGs were analyzed using the DAVID database [26], which included Kyoto Encyclopedia of Genes and Genomes (KEGG) and Gene Ontology (GO) enrichment analysis, with a significance threshold set at p $\leq$ 0.05. Subsequently, GO each subterms top 10 and KEGG pathways top 30 were selected for visualization purposes. The figures were generated utilizing an online platform for data analysis and visualization (https://www.bioinformatics. com.cn) [27].

## 2.4 Quantitative real-time polymerase chain reaction (qRT-PCR) analysis

We collected 5 ml peripheral blood samples from three spinal fracture patients within 7 days after SCI, who were admitted to Shangnan County Hospital in Shangnan County, Shaanxi Province, China and had no underlying diseases. Additionally, we obtained samples from three normal control subjects (spinal fracture without SCI). Detailed patient information is provided in Table 1. After collection, the peripheral blood samples were centrifuged at 1500 rpm for 15 minutes within 15 minutes of collection time. The leukocyte layer (interface layer) was carefully aspirated using a pipette and transferred into a tube containing 10 ml of red blood cell lysis buffer (BioLegend). Subsequently, the tube was incubated in darkness for 15 minutes followed by another centrifugation step at 1500 rpm for 10 minutes. The supernatant was discarded and the resulting cell pellet was resuspended in 1 ml TRIZOL (Ambion) for RNA extraction. Total RNA from leukocytes was extracted using the TRIZOL method with an average yield ranging from 15 to 25 μg per each collected sample of peripheral blood (5 ml). All extracted total RNA samples were stored in liquid nitrogen until further analysis. For cDNA synthesis, one microgram of total RNA from each sample was converted into cDNA using the First Strand cDNA Synthesis Kit (Takara Bio). The primers used in this study,

**Table 1. The detail clinical information of 6 samples.**

| Sample | Gender | Age | State | Injury location |
|--------|--------|-----|-------|-----------------|
| SCI+1 | female | 34 | spinal fracture with SCI | C5 vertebral fracture with SCI |
| SCI +2 | male | 50 | spinal fracture with SCI | C6 vertebral fracture with SCI |
| SCI +3 | male | 42 | spinal fracture with SCI | T12 vertebral fracture with SCI |
| Norm -1 | female | 35 | spinal fracture without SCI | T12 vertebral fracture without SCI |
| Norm -2 | female | 45 | spinal fracture without SCI | T10 vertebral fracture without SCI |
| Norm -3 | male | 40 | spinal fracture without SCI | T10 vertebral fracture without SCI |

including 9 SCI-DRGs and *GAPDH*, were designed and validated by Takara Bio Inc., as listed in S1 Table. For qRT-PCR analysis, the template cDNA (10 ng/rxn), SYBR Green fluorescent reagent from Takara Bio, and Roche LightCycler® 480 II were employed. The reaction conditions consisted of pre-denaturation at 95˚C for 5 minutes, denaturation at 95˚C for 30 seconds, annealing at 58˚C for 30 seconds, extension at 72˚C for 30 seconds, with a total of 45 cycles. *GAPDH* served as the reference control gene and relative gene expression was determined using the $2^{-\Delta\Delta CT}$ method. In our study, both the SCI group and the control group comprised three samples each. Moreover, all samples underwent three technical replicates during qRT-PCR analysis, and the results were subsequently averaged. This study was approved by the Ethics Committee of Shangnan County Hospital and conducted in accordance with the Helsinki Declaration. Written informed consent was obtained from all patients prior to their participation in the study. All data were de-identified and presented as mean ± standard deviation (SD). Statistical analysis was performed using GraphPad Prism software (version 9). Student's t-test was utilized to determine statistical significance, with $p < 0.05$ considered significant.

## 2.5 Identification of pivotal genes associated with SCI-induced disulfidoptosis in humans (hSCI-DRGs)

The intersection of SCI-DEGs and DRGs in the experimental cohort yielded a set of key genes, termed SCI-DRGs. The expression trends of these SCI-DRGs were validated in an independent dataset, followed by further validation using spatial transcriptomic data (GSE234774-GSM7474508). Finally, peripheral blood leukocyte samples from individuals with spinal cord injury were utilized to obtain a set of human-specific SCI-DRGs (hSCI-DRGs), which were subsequently employed for subsequent analysis. Conversion of gene symbol names between *human* and *mouse* species was performed using the "biomaRt, version 2.58.0 " package in R language.

## 2.6 Gene set enrichment analysis (GSEA) for KEGG pathways related to hSCI-DRGs

We employed the "c2.cp.kegg.v11.0.symbols" gene set from the Molecular Signatures Database (MSigDB, http://software.broadinstitute.org/gsea/msigdb) as the reference set for conducting GSEA-KEGG analysis on hSCI-DRGs to investigate their biological significance and functions [28]. Data analysis and visualization were performed using R software packages including "limma," "org.Hs.eg.db, version 3.2.0," "clusterProfiler, version 3.16.1," and "enrichplot, version 1.8.1," To ensure normalized enrichment scores during the analysis, we applied a corrected p-value threshold of < 0.05 and conducted 1000 permutations.

## 2.7 Immune infiltration analysis

CIBERSORT is an analysis tool specifically designed for evaluating the infiltration of immune cells [29]. By employing a reference dataset comprising 22 subtypes of immune cells and utilizing 1000 permutations to enhance accuracy, it enables estimation of immune cell abundance. When combined with the LM22 feature gene matrix, this process facilitates the selection of samples exhibiting a p-value less than 0.05, thereby obtaining an immune cell infiltration matrix. Subsequently, only data demonstrating an enrichment ratio greater than zero are retained for the immune cell infiltration matrix. Pearson correlation analysis is then employed to explore the association between infiltrating immune cells and immune gene signatures

within hSCI-DRGs. The R package was utilized for conducting the analysis on immune infiltration, while visualization was performed using "ggplot2" and "pheatmap" packages in R [30].

## 2.8 Transcription factor prediction and drug target prediction of hSCI-DRGs

The prediction of transcription factors for hSCI-DRGs was conducted using the online tool (https://jingle.shinyapps.io/TF_Target_Finder/) [31]. This website tool retrieved transcription factors specific to hSCI-DRGs from the hTFtarget, KnockTF, FIMO_JASPAR, and ENCODE databases by taking their intersection. Subsequently, correlation coefficients related to neural tissues in the Genotype-Tissue Expression Project (GTEx) database were calculated with a screening threshold set at greater than 0.3. Furthermore, Pathway and Process enrichment analysis of the predicted transcription factors was conducted using the online tool Metascape (https://metascape.org/gp/index.html#/main/step1) [32]. We selected hSCI-DRGs and their transcription factor-related targeted drugs through the DGIdb database (https://dgidb. genome.wustl.edu/), using default filtering criteria [33], and constructed a regulatory network comprising hSCI-DRGs, transcription factors, and drugs. Finally, Cytoscape software (version 3.7.2) was employed to visualize the analysis results [34].

## 3. Results

### 3.1 SCI-DRGs validation

To obtain the SCI-DRGs, we intersected 7387 SCI-DEGs (heatmap and volcano plot displayed in Fig 1A and 1B, SCI-DEGs listed in S2 Table, principal component analysis of SCI data set showed in S1 Fig) with 24 DRGs, and discovered a subset of 15 SCI-DRGs (*Flna*, *Ndufs1*, *Myh10*, *Iqgap1*, *Pdlim1*, *Slc3a2*, *Tln1*, *Actb*, *Nckap1*, *Actn4*, *Capzb*, *Flnb*, *Rpn1*, *Nubpl*, *Dstn*), as illustrated in Fig 1C. The expression heatmap of these 15 SCI-DRGs are depicted in Fig 1D.

### 3.2 Enrichment analysis of SCI-DEGs

To understand the biological functions of SCI-DEGs related genes, Gene Ontology (GO) and Kyoto Encyclopedia of Genes and Genomes (KEGG) analysis were performed. The 7387 SCI-DEGs were submitted to the DAVID database for GO and KEGG analysis. The GO analysis results revealed 1302 entries associated with SCI-DEGs (detailed results in S3 Table). Fig 2A presents the top ten entries in Biological Process (BP), Cellular Component (CC), and Molecular Function (MF) categories based on the GO enrichment analysis of SCI-DEGs. The KEGG analysis identified 185 pathways linked to SCI-DEGs (detailed results in S4 Table). Fig 2B illustrates the KEGG enrichment analysis top 30 outcomes of SCI-DEGs.

### 3.3 hSCI-DRGs validation

To acquire hSCI-DRGs, we conducted rigorous validation procedures utilizing online datasets, spatial transcriptome data from SCI samples, and RNA specimens obtained from peripheral blood leukocytes of human SCI patients. The expression levels of 15 SCI-DRGs were validated in the validation cohort dataset. Among them, 9 SCI-DRGs exhibited consistent expression levels with the experimental cohort (*Actb*, *Actn4*, *Capzb*, *Myh10*, *Nckap1*, *Ndufs1*, *Pdlim1*, *Slc3a2*, *Tln1*). Subsequently, these 9 SCI-DRGs were further confirmed in the mouse spatial transcriptome dataset GSE234774-GSM7474508 and demonstrated consistent expression trends with both the experimental and validation cohorts. Furthermore, qRT-PCR was performed to validate these 9 SCI-DRGs using peripheral blood leukocyte RNA samples from individuals with SCI. Ultimately, three hSCI-DRGs (*CAPZB*, *SLC3A2*,

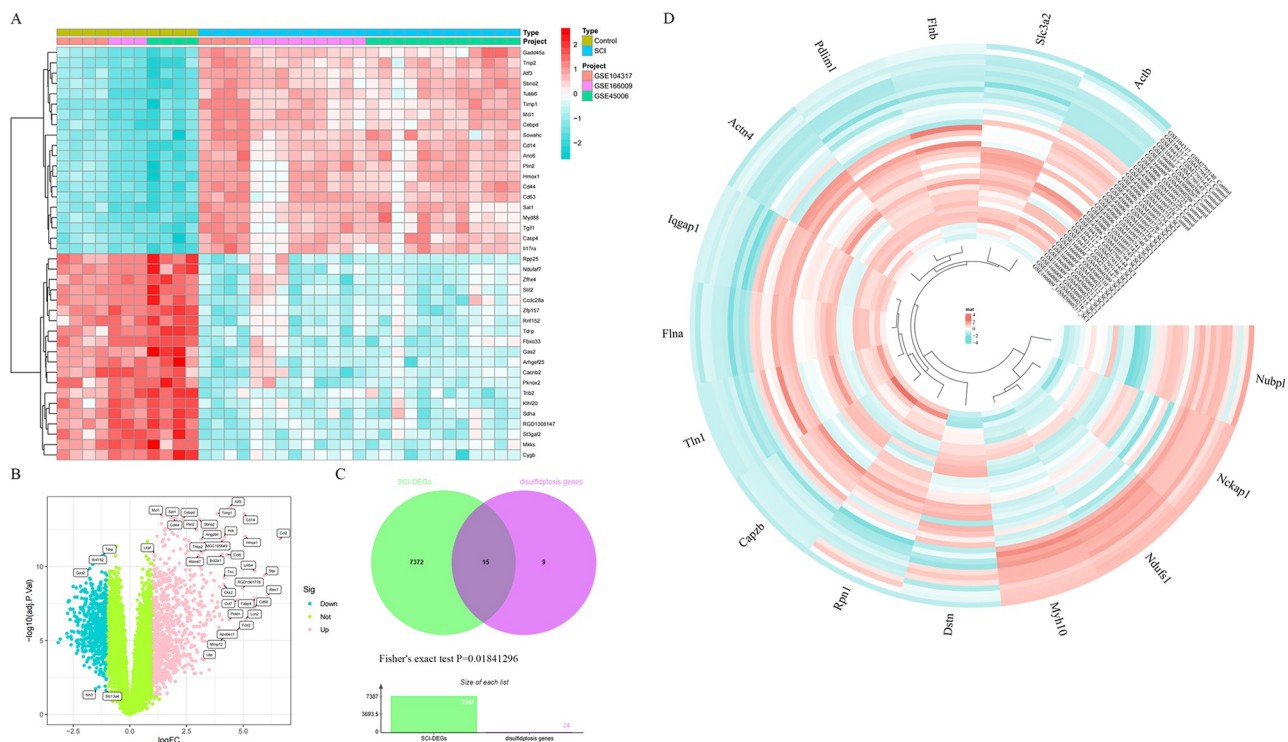

**Fig 1. Fig 1 display SCI-DEGs and SCI-DRGs.** Fig 1A heatmap shows the 20 upregulated and downregulated of 7387 SCI-DEGs genes with the most significant differences, the red color represents upregulated genes, the wathet color represents downregulated genes, and the distribution of differentially expressed genes in spinal cord injury, control group, and each dataset. Fig 1B highlights the significant DEGs associated with SCI in a volcano plot. Fig 1C illustrates the identification of SCI-DRGs, through the Fisher's exact test, the P value is 0.01841296. The two gene sets are not independent, that is, there is a significant correlation. Fig 1D shows the expression heatmap of 15 SCI-DRGs in each sample, the red color represents upregulated genes, the wathet color represents downregulated genes, and the distribution of differentially expressed genes in spinal cord injury, control group.

*TLN1*) were identified as shown in Fig 3. The expression levels and trends of *CAPZB*, *SCL3A2*, and *TLN1* in the experimental cohort, validation cohort, and spatial transcriptome are presented in Fig 4A–4C.

## 3.4 GSEA-KEGG analysis of hSCI-DRGs

To gain a clearer understanding of the potential mechanisms underlying disulfidptosis in SCI, we performed a single-gene GSEA-KEGG analysis on hSCI-DRGs to elucidate the roles and significance of key genes involved in disulfidptosis within SCI. Fig 5A–5F illustrates the top 10 pathways enriched with three hSCI-DRGs. Our comprehensive analysis reveals that *CAPZB* primarily contributes to cell cycle regulation, complement and coagulation cascades, lysosomal function, natural killer cell-mediated cytotoxicity, primary immunodeficiency, calcium signaling pathway modulation, long-term potentiation processes, neuroactive ligand receptor interactions, steroid biosynthesis as well as terpenoid backbone biosynthesis. *SLC3A2* is predominantly associated with cytokine-cytokine receptor interaction networks, MAPK signaling pathway activation, NOD-like receptor signaling pathway involvement, steroid biosynthesis regulation along with terpenoid backbone biosynthesis processes, B cell receptor signaling pathway participation, glycosaminoglycan degradation, lysosomal function, natural killer cell-mediated cytotoxicity and primary immunodeficiency. *TLN1* primarily participates

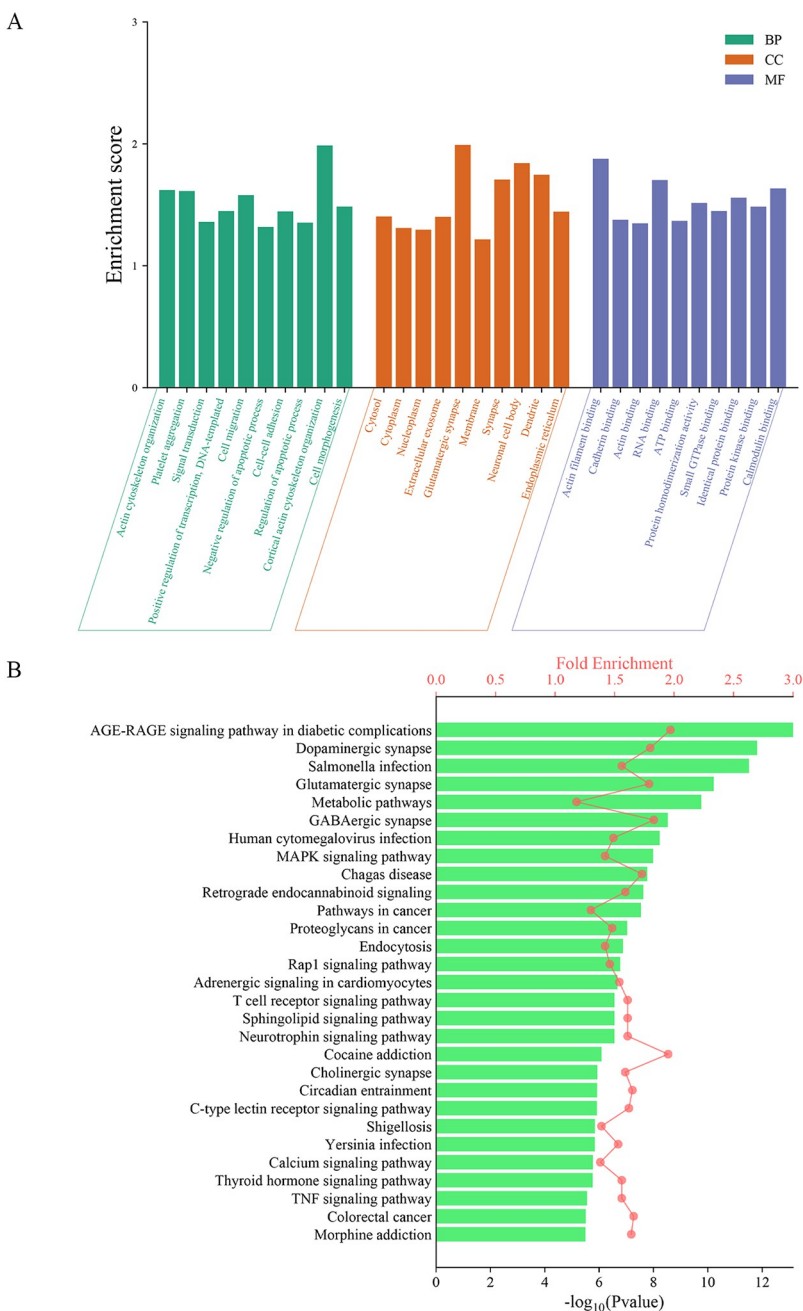

**Fig 2. Fig 2 presents the results of the GO and KEGG analyses of SCI-DEGs.** Fig 2A shows the GO analysis, displaying top 10 enriched terms (BP, CC, MF) related to SCI according to the fold enrichment,; detailed results can be found in S3 Table. Fig 2B represents the KEGG pathway analysis, revealing top 30 enriched pathways in SCI according to the fold enrichment and P Value, detailed results can be found in S4 Table.

in regulating the cell cycle progression dynamics while also being involved in complement and coagulation cascades modulation, lysosomal function maintenance, natural killer cell-mediated cytotoxicity promotion, primary immunodeficiency, development facilitation alongside calcium signaling pathway, modulation effects on long-term potentiation processes as well as neuroactive ligand receptor interactions.

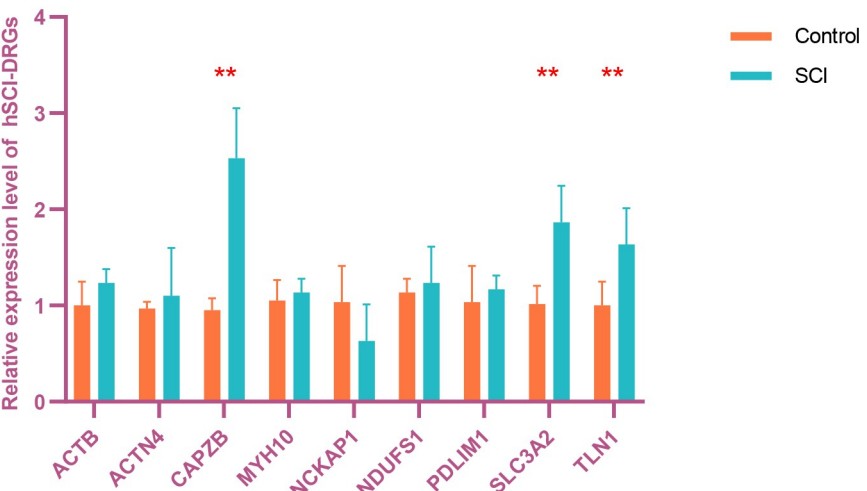

**Fig 3. Fig 3 displays the differential expression of *CAPZB*, *SCL3A2*, and *TLN1* in peripheral blood leukocytes within 7 days after SCI, as revealed by our qRT-PCR analysis in humans.** Ultimately, we observed that there were differential expressions of the three genes, *CAPZB*, *SCL3A2*, and *TLN1* and chosen *CAPZB*, *SCL3A2*, and *TLN1* for subsequent analysis in hSCI-DRGs. In the figure, * indicates a P-value less than or equal to 0.05, and ** indicates a P-value less than or equal to 0.01.

## 3.5 Immune infiltration analysis

To investigate the differences in the immune microenvironment between patients with SCI and the control group, we employed the CIBERSORT algorithm. As depicted in Fig 6A (detailed information presented in S5 Table), SCI samples exhibited a lower proportion of naive B cells compared to the control group, while activated NK cells were more abundant in SCI. Additionally, we examined the association between hSCI-DRGs and the immune micro-environment. We observed that *CAPZB* displayed a positive correlation with resting dendritic cells, resting mast cells, resting NK cells, CD8+ T cells, and a negative correlation with naive CD4+ T cells, naive B cells, activated dendritic cells, M1 macrophages, and activated mast cells as illustrated in Fig 6B. *SLC3A2* demonstrated a positive correlation with follicular helper T cells and a negative correlation with resting dendritic cells as shown in Fig 6C. *TLN1* exhibited a positive correlation with resting dendritic cells and resting mast cells but showed a negative correlation with follicular helper T-cells plasma cell population, activated dendritic cell population, activated mast cell population, M1 macrophage populations as illustrated in Fig 6D.

## 3.6 Construction of regulatory networks of transcription factors and drug targets for hSCI-DRGs

Through the TF_Target_Finder online tool search, we predict that *CAPZB*'s transcription factor is *STAT1*, *SLC3A2*'s transcription factor is *CREB1*, *JUND*, *SP1* and *STAT3*, and *TLN1*'s predicted transcription factor is *RELA* and *JUN*.. Pathway enrichment analysis using Metascape online tool revealed that these transcription factors are predominantly enriched in various signaling pathways including EGF EGFR signaling pathway, Oncostatin M signaling pathway, AGE RAGE pathway, Androgen receptor signaling pathway, Osteoclast differentiation pathway, Brain-derived neurotrophic factor BDNF signaling pathway, PID AP1 PATHWAY, Corticotropin-releasing hormone signaling pathway, and Non-genomic actions of 1, 25-dihydroxyvitamin D3 pathway (Fig 7A). Searching hSCI-DRGs in the DGIdb database

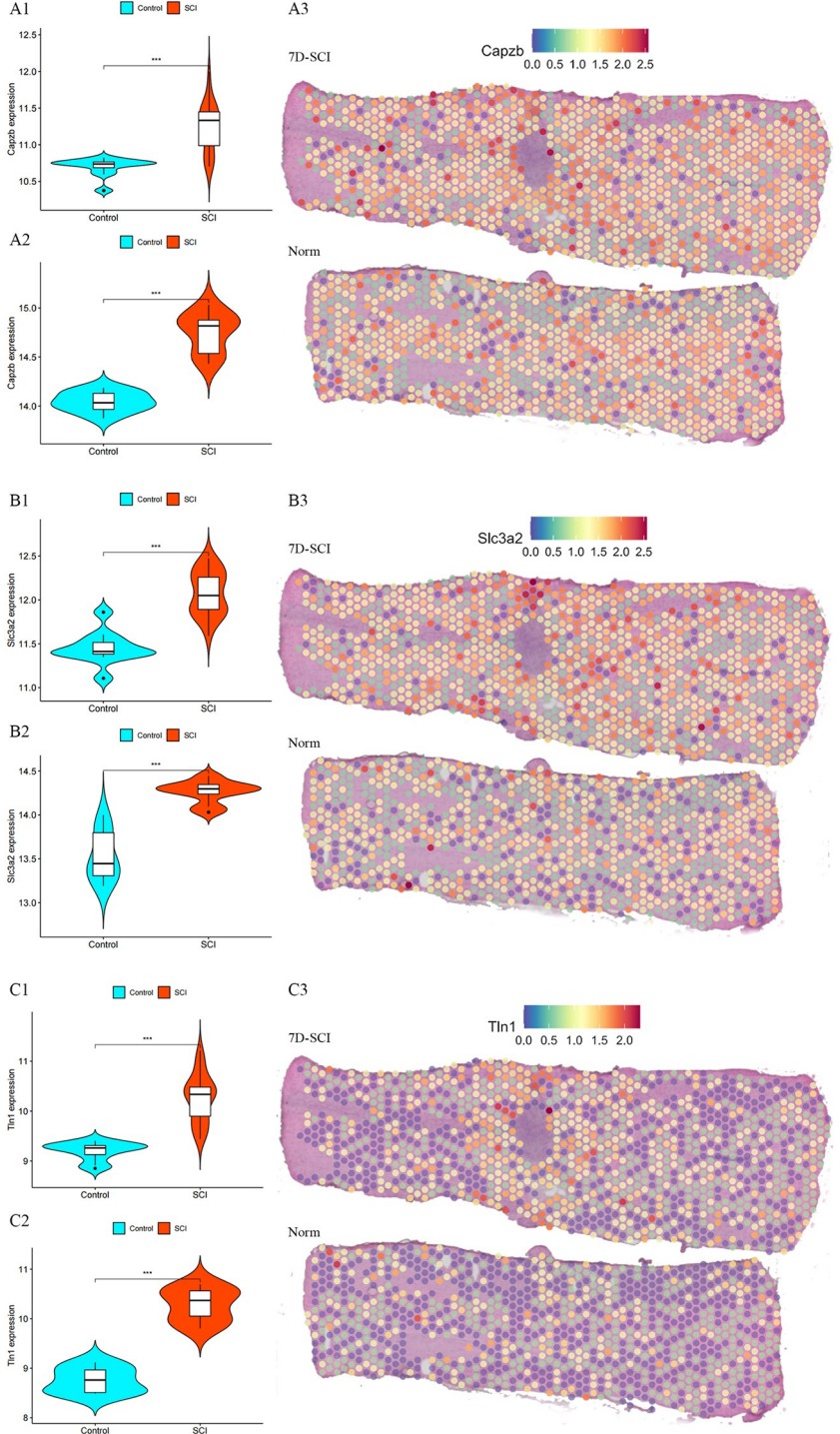

**Fig 4. Fig 4 depicts the expression profiles of hSCI-DRGs in the experimental cohort, validation cohort, and spatial transcriptomic data.** 7D-SCI represents the spatial transcriptomic expression profile data of mice 7 days after SCI, while Norm denotes the control group spatial transcriptomic data. Fig 4A shows the expression levels and trends of *Capzb* in the experimental cohort (A1), validation cohort (A2), and spatial transcriptomic data GSE234774-GSM7474508 (A3), both datasets show consistent expression trends, the spatial transcriptomics data show that the *Capzb* is expressed at higher levels in the tissue surrounding spinal cord injury lesions. Fig 4B shows the expression levels and trends of *Slc3a2* in the experimental cohort (B1), validation cohort (B2), and spatial transcriptomic data GSE234774-GSM7474508 (B3), both datasets show consistent expression trends, the spatial transcriptomics data show that the *Slc3a2* is expressed at higher levels in the tissue surrounding spinal cord injury lesions. Fig 4C shows the

expression levels and trends of *Tln1* in the experimental cohort (C1), validation cohort (C2), and spatial transcriptomic data GSE234774-GSM7474508 (C3), both datasets show consistent expression trends, the spatial transcriptomics data show that the *Tln1* is expressed at higher levels in the tissue surrounding spinal cord injury lesions.

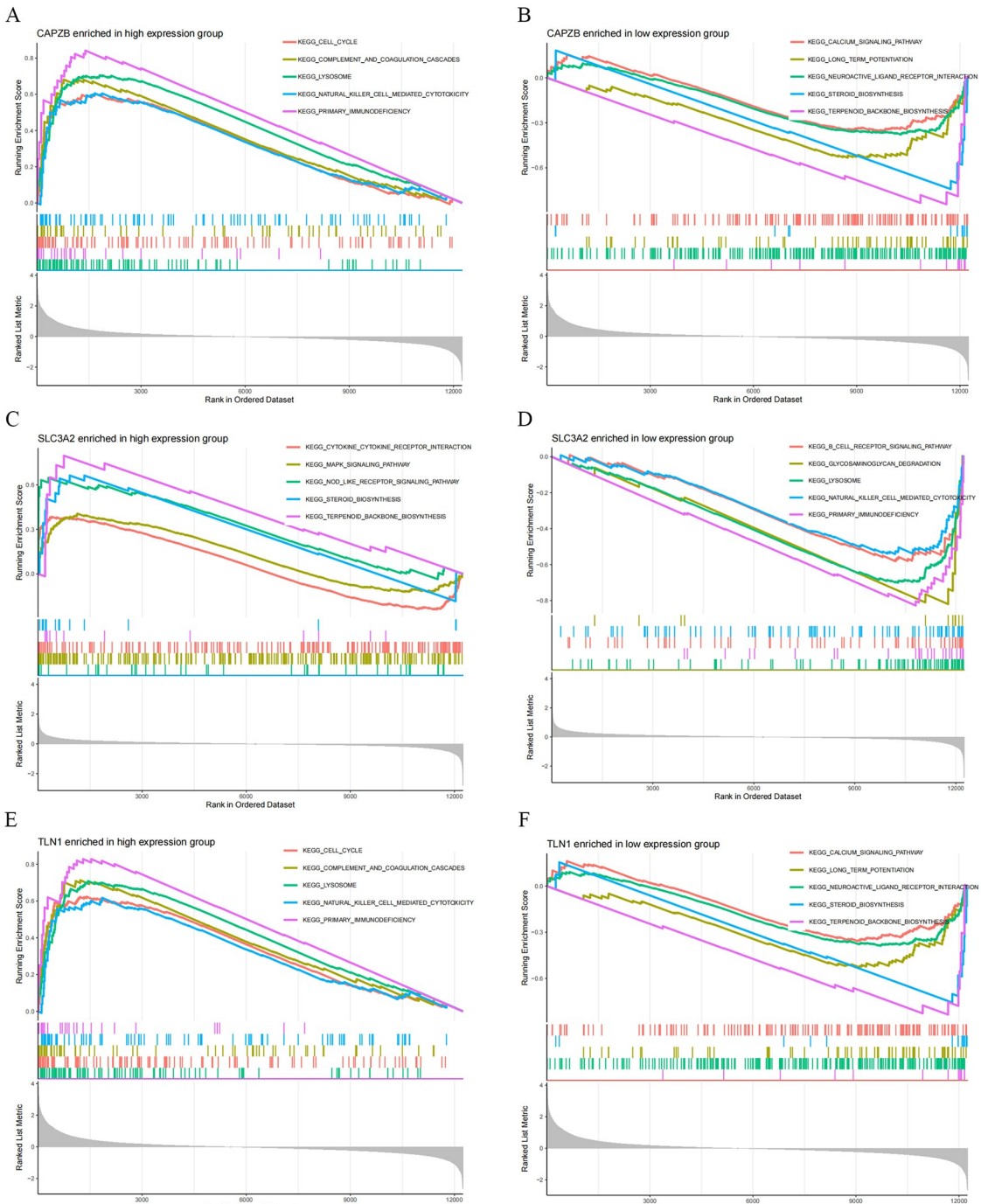

**Fig 5. Fig 5 shows the single-gene GSEA-KEGG analysis top 5 result in SCI group of hSCI-DRGs, and the P value of the presented pathways were all less than 0.05.** Fig 5A, 5C and 5E represents the enrichment analysis results of *CAPZB*, *SLC3A2* and *TLN1* in the high-expression group, while Fig 5B, 5D and 5F shows the enrichment analysis results of *CAPZB*, *SLC3A2* and *TLN1* in the low-expression group.

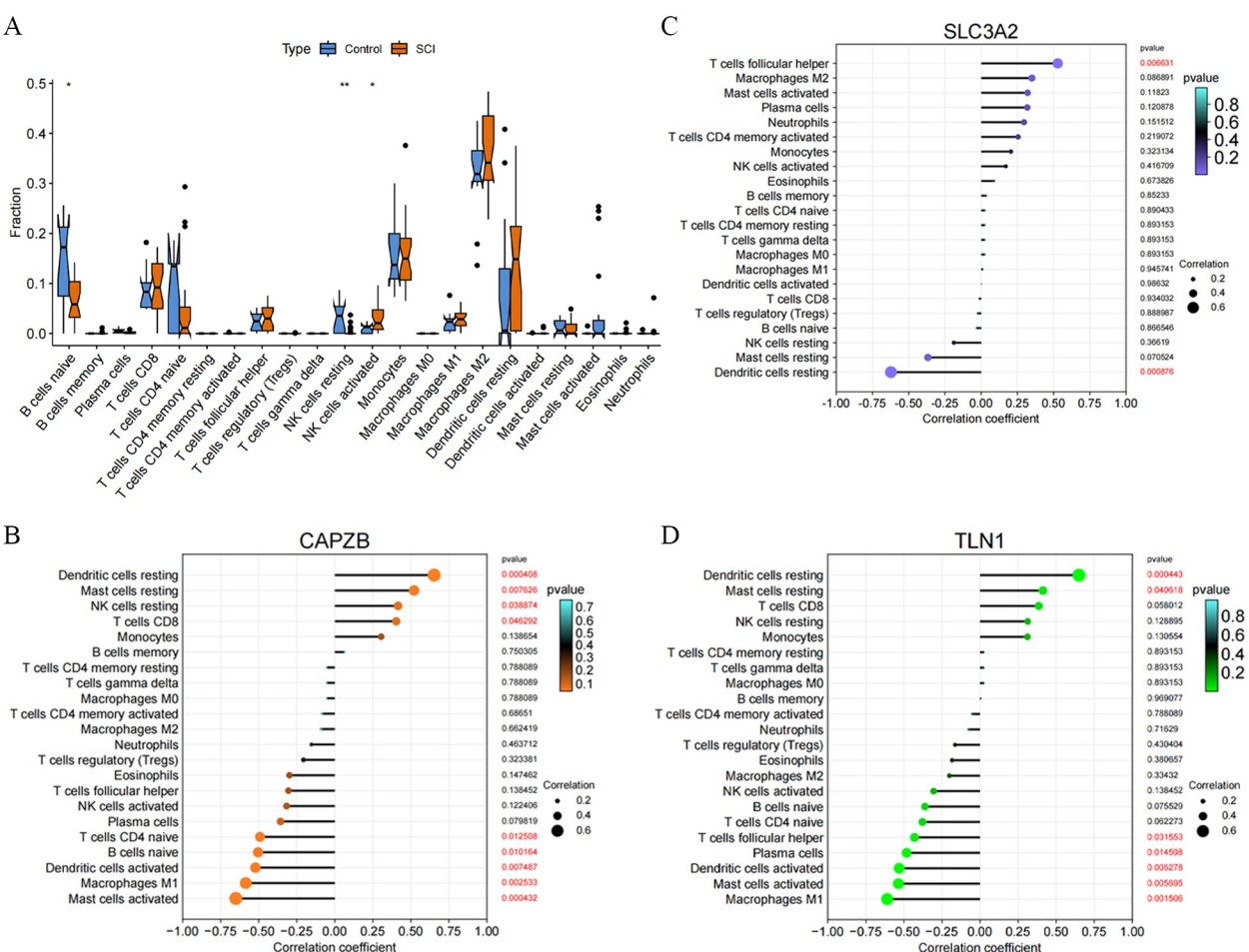

**Fig 6. Fig 6 presents immune cell infiltration analysis of hSCI-DRGs.** Fig 6A demonstrates the exploration of differences in the immune microenvironment between SCI and normal samples using the CIBERSORT algorithm, the results showed that hSCI-DRGs were associated with B cells naive, NK cells resting and NK cells activated, detailed results can be found in S5 Table. Fig 6B shows that *CAPZB* is positively correlated with T cell follicular helper and negatively correlated with Dendritic cells resting. Fig 6C, 6Dshows that *SLC3A2* and *TLN1* was positively correlated with Dendritic cells resting and Mast cells resting, and negatively correlated with Mast cells activated and Macrophages M1. Red font markers in the figure indicate a P-value less than or equal to 0.05.

yielded IGN523 as a targeted drug for *SLC3A2*; however, no interacting drugs were found for *CAPZB* and *TLN1*. By searching the transcription factors of hSCI-DRGs in the DGIdb database, a total of 140 target drugs were identified for these seven transcription factors with 43 already approved (S6 Table), as illustrated in Fig 7B.

## 4. Discussion

The incidence rate of SCI is characterized by its high frequency, substantial costs, significant disability rates, and early age of onset [35]. Severe SCI places considerable physical, psychological, and financial burdens on patients and their families. In China alone, there are a total of 759,302 traumatic SCI patients reported with an annual increase of 66,374 cases [36]. Furthermore, data from the United States reveals an annual incidence rate of approximately 17,000 individuals for SCI; the first-year cost for a patient with high tetraplegia exceeds 1 million US dollars [37]. Despite extensive investments in both financial and human resources towards SCI research globally [7, 8, 38], progress in treatment remains minimal.

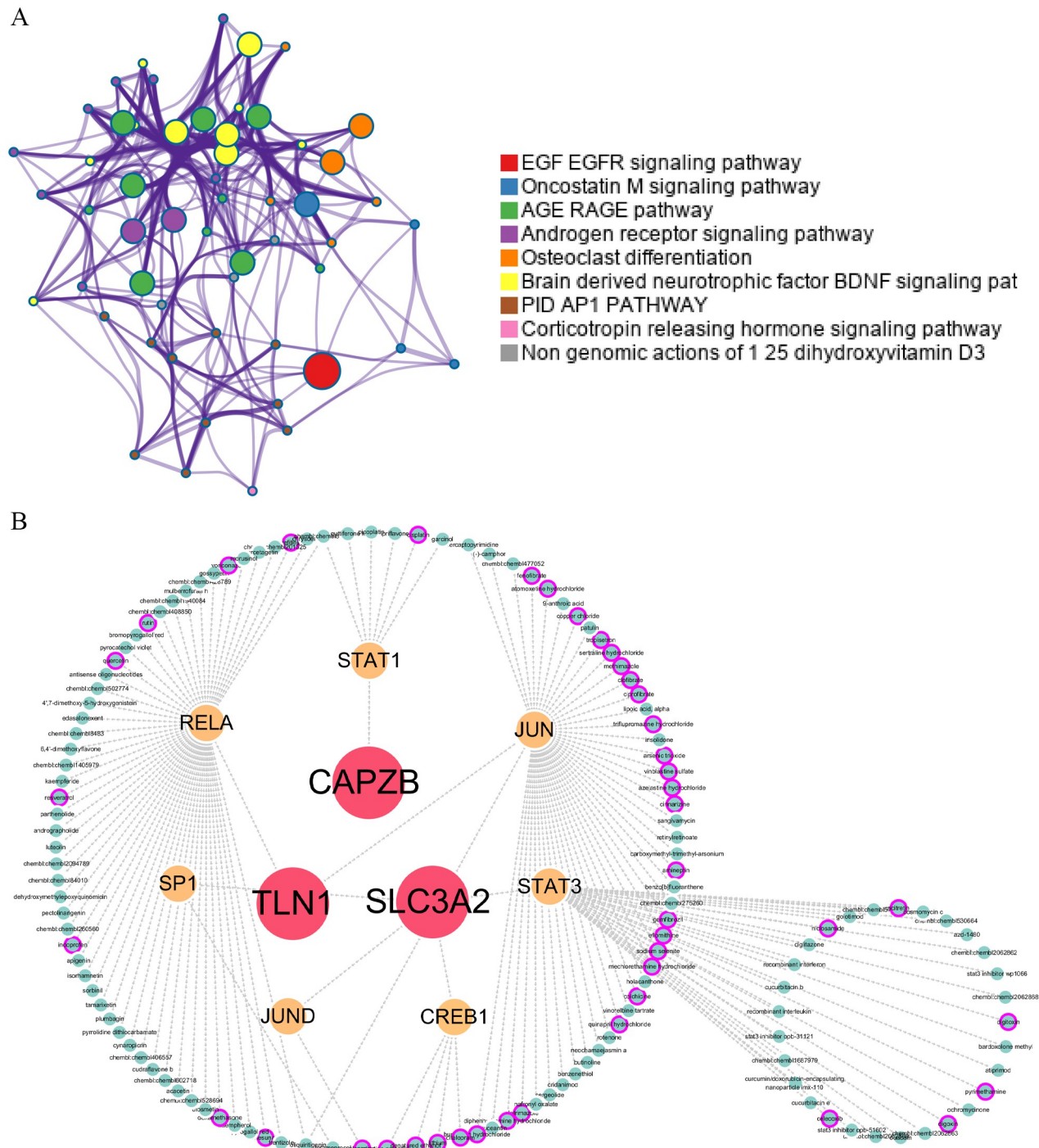

**Fig 7.** Fig 7A displays the transcription factor functional enrichment analysis of hSCI-DRGs. Through analysis, we retrieved seven transcription factors related to hSCI-DRGs and analyzed the regulatory pathways related to the transcription factors, detailed results can be found in S6 Table. Fig 7B shows the hSCI-DRGs-transcription factor-drug regulatory network and 140 transcription factor-related regulatory drugs. 43 of the drugs marked by purple circles have now been validated.

Currently, SCI's molecular biology research is primarily focused on PCD, encompassing apoptosis, necroptosis, autophagy, ferroptosis, pyroptosis, and paraptosis. Recently, disulfidptosis has emerged as a novel cell death pathway that has garnered increasing attention [17]. Disulfidptosis has been demonstrated to play a pivotal role in cancer [39–42], neurodegenerative diseases [43], and metabolic disorders [44]. Notably, the interplay between ferroptosis and disulfidptosis has been confirmed in current studies with *SLC7A11* playing a crucial role in this process [45, 46]. Ferroptosis-mediated cell death plays a predominant role in the pathogenesis of SCI [47], with acute-phase ferroptosis occurring within two days post-injury and subacute-phase occurring between three to fourteen days post-injury [48]. However, the involvement of disulfidptosis in spinal cord injury remains unexplored.

In this study, our focus was to investigate the regulatory mechanisms of disulfidptosis within the first 7 days following SCI. We successfully identified key genes associated with disulfidptosis in human SCI and performed functional enrichment analysis on these genes. Furthermore, we conducted an analysis of immune infiltration, and established comprehensive networks involving key genes, transcription factors, and potential drug targets.

In the analysis of SCI-DEGs using GO and KEGG, we identified significant results related toa variety of biological processes. The GO BP analysis revealed a strong focus on biological processes such as actin cytoskeleton organization, cortical actin cytoskeleton organization, and cell morphogenesis, displayed in Fig 2A. In terms of CC, the analysis highlighted extracellular exosome, cytosol, andnucleoplasm. Furthermore, MF was primarily centered around actin filament binding, cadherin binding, and calmodulin binding. These findings indicate that the GO analysis results of SCI-DEGs encompass numerous biological processes associated with actin collapse–a crucial process in disulfidptosis [17, 49, 50]. Regarding the KEGG analysis of SCI-DEGs, displayed in Fig 2B, several pathways are involved in the process after spinal cord injury, including: Dopaminergic synapse [51], Glutamatergic synapse [52], GABAergic synapse [51], MAPK signaling pathway [53], and Retrograde endocannabinoid signaling [51], which have been confirmed by previous studies. Additionally, based on KEGG enrichment analysis results, we observed associations between disulfidptosis and established diseases such as Proteoglycans in cancer [54], Non-alcoholic fatty liver disease [55, 56], Hepatocellular carcinoma [57, 58], Alzheimer's disease [43, 59], as well as lesser-studied diseases including Salmonella infection, Shigellosis Pathogenic Escherichia coli infection Amyotrophic lateral sclerosis.

After identification, we confirmed three hSCI-DRGs (*CAPZB*, *SLC3A2*, *TLN1*), displayed in Figs 3 and 4A–4C. *CAPZB* encodes the beta subunit of the barbed-end actin binding protein, which belongs to the F-actin capping protein family. Previous studies have demonstrated a significant increase in *b-actin*, *CAPG*, and *CAPZB* expression around the lesion site 14 days post-spinal cord injury [60]. Furthermore, *CAPZB* has been implicated in pathological processes associated with various diseases such as high-grade squamous epithelial lesions [61] and atherosclerosis [62]. *SLC3A2* is a member of the solute carrier family and encodes a cell surface transmembrane protein. Although no direct literature evidence supports its association with SCI, research has also revealed dysregulation of *Slc7a5* and its binding partner *Slc3a2* in chronic neuropathic pain conditions where they can serve as potential therapeutic targets [63]. *TLN1* encodes a cytoskeletal protein that localizes to areas of cell-substratum and cell-cell contacts. Currently, there is no research confirming its association with SCI. Other studies focus on elucidating how *TLN1* promotes tumor cell invasiveness, proliferation, and metastasis [64].

Through GSEA-KEGG analysis of hSCI-DRGs, we identified *CAPZB* and *TLN1* as primarily involved in similar pathways, displayed in Fig 5A, 5B, 5E and 5F. Recent studies have confirmed the involvement of disulfidptosis in biological processes related to lung adenocarcinoma and pancreatic cancer through the cell cycle pathway [65, 66]. Mounting evidence suggests that spinal cord injury activates the cell cycle, and inhibition of key cell cycle

regulatory pathways can mitigate injury-induced cell death, as well as oligodendrocyte precursor cells and astrocytes proliferation both in vitro and in vivo [67]. The complement and coagulation cascades participate in regulating acute to subacute neuroinflammation in a rat spinal cord contusion model [68]. Previous research has demonstrated that genes *Rac2*, *Itgb2*, and *Tyrobp* play crucial roles in SCI development via the "natural killer cell mediated cytotoxicity" pathway [69]. In the enrichment results of *SLC3A2*, displayed in Fig 5C and 5D, it has been demonstrated that the cytokine-cytokine receptor interaction pathway is implicated in immune infiltration in ischemic stroke [70], as well as disulfidptosis-related biological processes in ankylosing spondylitis and inflammatory bowel disease [71]. Given the shared pathophysiological mechanisms between spinal cord injury and ischemic stroke, such as neuronal apoptosis, neuroinflammation, oxidative stress, blood-brain barrier dysfunction, astrocyte proliferation, and axon growth inhibition [72], it is plausible to infer that the cytokine-cytokine receptor interaction pathway may also play a role in the pathophysiological processes of disulfidptosis following spinal cord injury. Furthermore, previous studies have indicated involvement of the MAPK signaling pathway [73], steroid biosynthesis [74], glycosaminoglycan degradation [75], and natural killer cell mediated cytotoxicity [69] pathways in the pathophysiology of SCI. Therefore, we hypothesize that *CAPZB* and *TLN1* may be involved in the pathological process of disulfidptosis following spinal cord injurythrough the cell cycle, coagulation cascades, and natural killer cell mediated cytotoxicity pathways. On the other hand, *SLC3A2* may participates in the process of disulfidptosis after SCI through cytokine-cytokine receptor interaction pathway, MAPK signaling pathway, steroid biosynthesis, glycosaminoglycan degradation, and natural killer cell mediated cytotoxicity pathways.

Through immune infiltration analysis, we observed a decreased proportion of naive B cells in SCI compared to the control group, while activated NK cells were found to be more abundant in SCI, displayed in Fig 6A [76, 77]. Additionally, we identified a close association between hSCI-DRGs and immune cells as mentioned earlier, displayed in Fig 6B–6D. Consistent with our findings, previous studies have also validated our analysis results. For instance, one study demonstrated the accumulation of *CAPZB* with immune cells at the injury center on day 14 post-SCI, with most reactive astrocytes located around the lesion center [60]. Another research revealed shared transcriptional features between in vitro activated plasmacytoid dendritic cells and those present in lupus nephritis lesions including *SLC7A5*, *SLC3A2*, *SLC7A11* along with ectonucleotide pyrophosphatase/phosphodiesterase 2 (*ENPP2*), where *SLC3A2* was implicated in the immune response in lupus nephritis [78]. Furthermore, it has been confirmed that efficient uptake of glutamine and leucine through *SLC1A5*, *SLC7A5*, and *SLC3A2* can enhance metabolic capacity and effector functions of NK and T cells targeting tumors for improved tumor recognition and infiltration [79, 80]. In a recent study focused on ischemic stroke diagnosis and immune biomarkers identification associated with T cell helper function, *TLN1* was discovered as an important marker [81]. Therefore, we infer that hSCI-DRGs play a crucial role in modulating the immune responses of CD8 T cells, NK cells, dendritic cells, and mast cells subsequent to SCI. Although comprehensive investigation into the immune-related molecular biology mechanisms of hSCI-DRGs in SCI is currently lacking attention from researchers but it is anticipated that this aspect will garner more interest in future studies.

We utilized the online tool TF_Target_Finder to identify transcription factors that target hSCI-DRGs and identified 7 candidates meeting our criteria. Subsequently, pathway enrichment analysis was performed for these 7 transcription factors, displayed in Fig 7A. Among them, the EGF-EGFR signaling pathway is implicated in cell proliferation, survival, and differentiation, its dysregulation has been linked to various cancers [82]. Furthermore, studies have demonstrated that the HB-EGF/EGFR pathway promotes neuroblast proliferation and sensory

neuron regeneration following olfactory epithelial injury in zebrafish [82]. The Oncostatin M signaling pathway and AGE-RAGE pathway are involved in signaling cascades related to inflammation, oxidative stress, and cellular dysfunction. The Brain-derived neurotrophic factor (BDNF) signaling pathway plays a role in neuronal survival, growth, and synaptic plasticity; interventions targeting this pathway have shown promise for improving cognitive decline in Alzheimer's disease patients [83]. The PID AP1 PATHWAY regulates biological processes such as cell proliferation, apoptosis, and inflammation. Through pathway enrichment analysis of hSCI-DRGs transcription factors, we discovered that 7 transcription factors related to hSCI-DRGs play a crucial role in cell survival and differentiation, neural function recovery, neural inflammation, and oxidative stress after SCI. In our search of the DGIdb database for hSCI-DRGs we found that *SLC3A2* is targeted by IGN523—a humanized antibody against CD98 currently used in anti-tumor applications [84], but its role in disulfidptosis remains unvalidated. Inputting the 7 transcription factors into the DGIdb database yielded 140 target drugs with 43 approved agents among them, displayed in Fig 7B. These approved or investigational drugs may hold potential as future treatments for disulfidptosis. Additionally, there are ongoing efforts towards developing targeted therapeutic agents directly affecting genes involved in disulfidptosis.

The limitations of this study are as follows: Firstly, we employed bioinformatics analysis and qRT-PCR methods exclusively to investigate key disulfidptosis genes following human SCI. Secondly, our focus was primarily on the occurrence and progression of disulfidptosis within 7 days post-SCI, leaving the expression patterns of disulfidptosis genes during other time periods unknown. Moreover, due to the intricate nature of neural tissues, there exist variations in the expression data of disulfidptosis genes across different neural tissues. In future studies, we intend to integrate single-cell data to further elucidate the expression and impact of disulfidptosis genes in distinct cell types after SCI. Given the challenges associated with obtaining human SCI tissue samples, we plan to validate our findings through animal experiments while also screening small molecule compounds that modulate disulfidptosis.

Strengths of this study include the following aspects. Firstly, we are the first observation in rat, mouse, and human peripheral blood leukocyte samples that disulfidptosis may involve in the pathophysiological processes following spinal cord injury. Secondly, by utilizing multiple online datasets, spatial transcriptomic data, and validating the expression levels and trends of disulfidptosis-related genes in human peripheral blood leukocytes, our results are reliable and have reference value..

## 5. Conclusions

In summary, the pathophysiological process following spinal cord injury probably involves disulfidptosis, and a total of three hSCI-DRGs (*CAPZB*, *SCL3A2*, and *TLN1*) have been filtered. Furthermore, all hSCI-DRGs may be implicated in post-SCI pathway regulation, immune infiltration, gene-transcription factor-drug regulatory network. These findings provide novel insights into understanding the pathological mechanisms underlying disulfidptosis in SCI and have implications for its diagnosis, treatment, and future research endeavors.

## Supporting information

**S1 Fig. Principal component analysis of SCI data set after batch correction.**
(JPG)

**S1 Table. Primers used in this study.**
(XLSX)

**S2 Table. Differentially expressed genes in SCI.**
(XLSX)

**S3 Table. GO analysis results of SCI-DEGs.**
(XLSX)

**S4 Table. KEGG analysis results of SCI-DEGs.**
(XLSX)

**S5 Table. Detailed immune microenvironment in SCI.**
(XLSX)

**S6 Table. Detailed regulatory networks of hSCI-DRGs-transcription factors-drug targets.**
(XLSX)

## Acknowledgments

We thank Mingjie Chen (Shanghai NewCore Biotechnology Co., Ltd.) for providing data analysis and visualization support.

## Author Contributions

**Data curation:** Shuang Wang, Jun Tian.

**Investigation:** Shuang Wang, Jun Tian, Shuling Zhang.

**Methodology:** Shuang Wang, Xinhua Liu.

**Project administration:** Shuling Zhang.

**Resources:** Jichun Yang.

**Software:** Shuang Wang, Jun Tian.

**Supervision:** Sizhu Liu, Lianwei Ke, Hongying He, Chaojiang Shang, Jichun Yang.

**Validation:** Sizhu Liu, Lianwei Ke, Hongying He, Chaojiang Shang.

**Visualization:** Jun Tian.

**Writing – original draft:** Shuang Wang, Jun Tian.

**Writing – review & editing:** Xinhua Liu, Jun Tian, Hongying He.

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
