## [Decision Letter · Decision Letter 0]

29 Nov 2024

PONE-D-24-34139Bioinformatics analysis of genes associated with disulfidptosis in spinal cord injuryPLOS ONE

Dear Dr. Tian,

Thank you for submitting your manuscript to PLOS ONE. After careful consideration, we feel that it has merit but does not fully meet PLOS ONE’s publication criteria as it currently stands. Therefore, we invite you to submit a revised version of the manuscript that addresses the points raised during the review process.

We look forward to receiving your revised manuscript.

Kind regards,

Zhiwen Luo

Academic Editor

PLOS ONE

Additional Editor Comments:

Thank you for submitting your manuscript to the Journal and as voucan see that the reviewer think your manuscript is interesting and provide valuable comments for your reference. Please submit the revised manuscript ASAP and also include a rebuttal that would clearly list all the responses to the reviewer's comments.

Reviewers' comments:

Reviewer's Responses to Questions

**Comments to the Author**

1. Is the manuscript technically sound, and do the data support the conclusions?

Reviewer #1: Partly

Reviewer #2: Yes

2. Has the statistical analysis been performed appropriately and rigorously? 

Reviewer #1: Yes

Reviewer #2: Yes

3. Have the authors made all data underlying the findings in their manuscript fully available?

Reviewer #1: No

Reviewer #2: Yes

4. Is the manuscript presented in an intelligible fashion and written in standard English?

Reviewer #1: Yes

Reviewer #2: Yes

5. Review Comments to the Author

Reviewer #1: Comments:

The paper performs bioinformatics analyses of transcriptomic data to assess differences in gene expression following spinal cord injury, with a particular focus on disulfidptosis-related genes. The study employed datasets from multiple species and used human patient-derived SCI and control samples to support their findings. The authors utilised online databases for deeper investigations of their genes of interest. Overall, the authors frequently reach conclusions that are not supported by the paper's findings and tend to overinterpret their analyses. In particular, the origin of the list of disulfidptosis-related genes that is relied upon very heavily in the paper is not described. More detailed information on the settings of bioinformatic tools used in the work are also required in order to accurately assess the appropriateness of much of the analysis. I recommend that this article is not accepted for publication in its current state. Very major revisions should be made or the article should be rejected.

Methods:

The methods should contain more information on how exactly batch effects were accounted for, and how the differential expression was performed. What settings were used in each of the programs, which versions, and what design formula was used for the differential expression analysis. More details are needed for how almost all of the bioinformatics tools were used.

Results:

1.For the differential expression analysis, it would be good to see a histogram of p-values to know that the test used is appropriate for the data. This could be put in the supplementary data.

2.It would also be informative to see a principal component analysis plot for the different samples shown in Figure 1. This could also go in supplementary data.

3.How were the genes chosen in Figure 1A? Is it the top 20 up- and downregulated genes? If so, this should be stated.

4.The bar chart in figure 1C should not have an arrow on the X axis, as this axis is showing discrete categories, not a continuous variable. This information is also provided in the Venn diagram, so may not be necessary to include. It would be better if the sizes of the circles in the Venn diagram were plotted relative to the size of each group. Furthermore, a statistical test like Fisher’s exact test should be performed to determine if this overlap is more than expected by chance.

5.Where was the list of DRGs obtained? I couldn’t find anything in the text regarding this. Was it from previous literature? If so, that needs to be accurately cited. This is an essential part that is missing from this paper.

6.Figure 1D is very hard to understand. The connected lines convey that there is a connection between individual samples, which as I understand, there is not. It would be better to plot this as a heatmap.

7.The figures are in too low resolution to clearly read gene names and gene ontologies.

8.In Figure 2, it would be better to plot an enrichment score rather than the number of genes. Given that we aren’t given information on the number of genes that make up each GO term, the number of genes that overlap here could be not very relevant if the GO term lists are of very different sizes.

9.In figure 3, it is best practice to show the control first on the plot, and the injury sample after.

10.In line 282, it should state that STAT1 was predicted as a transcription factor targeting CAPZB, not “the transcription factor for CAPZB”. The evidence in the paper is not sufficient to claim that STAT1 is a direct transcriptional regulator of CAPZB. Same goes for the rest of the transcription factors mentioned here. The authors should also be careful to double-check for italicizing of gene names.

11.Most of these are quite general transcription factors, with many global effects so the authors should write cautiously about using them as drug targets. Futhermore, more empirical/experimental evidence is required to demonstrate that the identified SCI-DRGs are actually involved in the pathology of SCI before they could be considered as drug targets.

12.For the ceRNA analysis, the authors should perform co-expression analysis of the protein-coding mRNAs, lncRNAs and miRNAs and integrate these results with the database search they have performed.

Discussion:

13.In line 374-376 of the discussion, the authors make conclusions that are not supported by the data of the paper.

14.Line 444-447, they are also making claims that are not supported by the data of the paper. The authors should not use the word “confirm” based only on bioinformatic analysis of transcriptomic data.

15.Line 453-454, the data of this paper does not provide sufficient evidence for this claim.

Reviewer #2: I believe this manuscript meets the publication requirements of PLOS ONE. Therefore, I recommend its acceptance pending minor revisions. Below are my comments for the authors:

1) The figure legends should be reworded. I suggest assigning clear titles to each figure and describing each panel separately.

2) Once a term is abbreviated, use only the abbreviation in subsequent mentions (examples: lines 47–56, 66–100, 38–79).

3) I recommend adding a sentence at the end of the last paragraph of the introduction to summarize the major takeaway of the study.

4) The section 2.4 is too dense and wordy. Consider renaming it “RNA extraction and qRT-PCR,” and move the latter part into a new subsection (e.g., 2.5).

5) Provide details about the technical and biological replicates used in the qRT-PCR experiments.

6) Sections 3.1, 3.2, 3.3 are overly descriptive. At the beginning of each paragraph, include a concise rationale explaining why the analysis was performed. Conclude each paragraph with a brief summary of the findings.

7) What does the literature say about the roles of CAPZB and SLC3A2? (lines 250–254).

8) Double-check punctuation (e.g., line 278).

9) When repeating your findings in the discussion, cite the corresponding figures whenever possible (examples: lines 336–341, 353–354).

10) Lines 376–379: Is this observation a finding of the current study or derived from the literature? In either case, cite the appropriate figures or references.

6. PLOS authors have the option to publish the peer review history of their article (what does this mean?). If published, this will include your full peer review and any attached files.

Reviewer #1: No

Reviewer #2: No

---

## [Author Response · Author response to Decision Letter 0]

14 Dec 2024

Editor Comments

Dear editor,

We have once again compared the template and revised the formatting of the manuscript.

Best regards.

Dear editor,

We have applied for an ORCID iD for the corresponding author and have updated the relevant information.

Best regards.

Dear editor,

Except for the ethics statement in the Methods section, the rest have been deleted.

Best regards.

Dear editor,

At the end of the manuscript, we have added a Supporting Information section, standardized the naming, and updated the citations in the text.

Best regards.

5. While revising your submission, please upload your figure files to the Preflight Analysis and Conversion Engine (PACE) digital diagnostic tool, https://pacev2.apexcovantage.com/. PACE helps ensure that figures meet PLOS requirements.

Dear Editor, 

we have uploaded all the figures in the manuscript to the PACE diagnostic system. After diagnosis and modification, they all meet the publication requirements of PLOS One.

Best regards.

Comments to the Author

Dear editor,

We hereby declare that our manuscript has not been published in any other journal, and our research has been approved by the Ethics Committee of Shangnan County Hospital. All authors have agreed to the publication of the article.

Best regards.

Answer questions to Reviewer 1

Comments:

The paper performs bioinformatics analyses of transcriptomic data to assess differences in gene expression following spinal cord injury, with a particular focus on disulfidptosis-related genes. The study employed datasets from multiple species and used human patient-derived SCI and control samples to support their findings. The authors utilised online databases for deeper investigations of their genes of interest. Overall, the authors frequently reach conclusions that are not supported by the paper's findings and tend to overinterpret their analyses. In particular, the origin of the list of disulfidptosis-related genes that is relied upon very heavily in the paper is not described. More detailed information on the settings of bioinformatic tools used in the work are also required in order to accurately assess the appropriateness of much of the analysis. I recommend that this article is not accepted for publication in its current state. Very major revisions should be made or the article should be rejected.

Dear reviewer,

Thank you for providing very unique and precise suggestions for this manuscript. We have made significant improvements to the manuscript based on your recommendations. We also hope that the manuscript meets the publication requirements of PLOS ONE. Thank you once again for your valuable suggestions.

Best regards.

Methods:

The methods should contain more information on how exactly batch effects were accounted for, and how the differential expression was performed. What settings were used in each of the programs, which versions, and what design formula was used for the differential expression analysis. More details are needed for how almost all of the bioinformatics tools were used.

Dear reviewer,

Thank you very much for pointing out the shortcomings of the article. In the Methods section, the author team has detailed how to remove batch effects and conduct differential analysis based on the issues you raised. Additionally, we have added information on the program settings and each program version. 

Best regards.

Results:

1. For the differential expression analysis, it would be good to see a histogram of p-values to know that the test used is appropriate for the data. This could be put in the supplementary data.

Dear reviewer,

In Table S2-5, detailed results of differential analysis, GO, KEGG, and immune infiltration analysis are presented, including detailed P or adjusted P values, as well as other detailed information. Additionally, we have added data bars for the P or adjusted P values in Table S2-5 to indicate their magnitude.

Best regards.

2. It would also be informative to see a principal component analysis plot for the different samples shown in Figure 1. This could also go in supplementary data.

Dear reviewer,

Thank you for pointing out our oversight. We have submitted the batch-corrected principal component analysis results in the supplementary materials, presented in S1 Figure.

Best regards.

3. How were the genes chosen in Figure 1A? Is it the top 20 up- and downregulated genes? If so, this should be stated.

Dear reviewer,

Thank you for pointing out our oversight. The top 20 significantly upregulated and downregulated genes are displayed in the heatmap in Fig 1A. We have added relevant information to the figure caption. 

Best regards.

4. The bar chart in figure 1C should not have an arrow on the X axis, as this axis is showing discrete categories, not a continuous variable. This information is also provided in the Venn diagram, so may not be necessary to include. It would be better if the sizes of the circles in the Venn diagram were plotted relative to the size of each group. Furthermore, a statistical test like Fisher’s exact test should be performed to determine if this overlap is more than expected by chance.

Dear reviewer,

Thank you for pointing out the shortcomings in the figures. We have made the necessary modifications to the corresponding images. Due to the significant disparity in the number of genes between the two sets, representing each group's size with circle sizes appeared highly incongruous. We conducted a Fisher's exact test on the Venn diagram results, yielding a p-value of 0.01841296, indicating that the two gene sets are not independent and exhibit significant correlation. Additionally, we have embedded the p-value into the Venn diagram.

Best regards.

5. Where was the list of DRGs obtained? I couldn’t find anything in the text regarding this. Was it from previous literature? If so, that needs to be accurately cited. This is an essential part that is missing from this paper.

Dear reviewer,

Thank you for pointing out the shortcomings in the paper. At the end of the Data Sources section in Section 2.1 of the manuscript, we have described that 24 differentially regulated genes (DRGs) originated from previous studies, and we have cited the relevant articles. In this revision, we have supplemented detailed information on these 24 DRGs.

Best regards.

6. Figure 1D is very hard to understand. The connected lines convey that there is a connection between individual samples, which as I understand, there is not. It would be better to plot this as a heatmap.

Dear reviewer,

Thank you for pointing out the shortcomings in the paper. In response to your suggestion, we have replaced the line graph with a heatmap, as shown in Fig 1D.

Best regards.

7. The figures are in too low resolution to clearly read gene names and gene ontologies.

Dear reviewer,

Thank you for pointing out the shortcomings in the manuscript. We have now replaced all the images in the document with the highest resolution TIFF versions. Simultaneously, we have uploaded the diagnostic system of PACE to PLOS One, and after modifications, all images now meet the requirements.

Best regards.

8. In Figure 2, it would be better to plot an enrichment score rather than the number of genes. Given that we aren’t given information on the number of genes that make up each GO term, the number of genes that overlap here could be not very relevant if the GO term lists are of very different sizes.

Dear reviewer,

Thank you for pointing out the shortcomings in the paper. Following your advice, we have redrawn the GO and KEGG-related images based on the enrichment scores. Additionally, we have updated the relevant sections and references in the manuscript accordingly.

Best regards.

9. In figure 3, it is best practice to show the control first on the plot, and the injury sample after.

Dear Reviewer,

Thank you for highlighting the shortcomings of the paper. Following your guidance, we have swapped the positions of the damaged samples and the control group samples.

Best regards.

10. In line 282, it should state that STAT1 was predicted as a transcription factor targeting CAPZB, not “the transcription factor for CAPZB”. The evidence in the paper is not sufficient to claim that STAT1 is a direct transcriptional regulator of CAPZB. Same goes for the rest of the transcription factors mentioned here. The authors should also be careful to double-check for italicizing of gene names.

Dear Reviewer,

Thank you for pointing out the shortcomings of the paper. We have made changes to the description of transcription factor prediction in the manuscript, and we have also adjusted the italicization of gene names as suggested.

Best regards.

11. Most of these are quite general transcription factors, with many global effects so the authors should write cautiously about using them as drug targets. Futhermore, more empirical/experimental evidence is required to demonstrate that the identified SCI-DRGs are actually involved in the pathology of SCI before they could be considered as drug targets.

Dear Reviewer,

Thank you for your guidance on the content of the sections. When designing the analytical approach in our study, we took into consideration the broad biological processes following spinal cord injury, with transcription factors playing a crucial role. While the research results may not directly serve as a basis for drug targeting, they have provided us with a new avenue for future investigations. If this approach is validated, we plan to cautiously verify these as potential therapeutic targets through experiments or explore new methods to regulate these transcription factors in local tissues.

Best regards.

12. For the ceRNA analysis, the authors should perform co-expression analysis of the protein-coding mRNAs, lncRNAs and miRNAs and integrate these results with the database search they have performed.

Dear Reviewer,

Thank you for pointing out the shortcomings of the paper. Following your advice, we conducted another search in existing databases but did not find human spinal cord injury-related miRNA and lncRNA data suitable for validation. Regarding the predicted human gene ceRNA network data in the manuscript, considering the lack of necessary data for co-expression analysis and integration, after careful consideration by the authors, we have unanimously decided to remove the sections related to the ceRNA network from the manuscript. Once again, we appreciate your suggestions, and we sincerely apologize for any inconvenience caused by the contentious parts of the paper.

Best regards.

Discussion:

13. In line 374-376 of the discussion, the authors make conclusions that are not supported by the data of the paper.

Dear Reviewer, 

Thank you for pointing out the shortcomings of the paper. We have revised the relevant statements based on your feedback. The modified text is as follows: "Therefore, we hypothesize that CAPZB and TLN1 may be involved in the pathological process of disulfidptosis following spinal cord injury through the cell cycle, coagulation cascades, and natural killer cell-mediated cytotoxicity pathways. On the other hand, SLC3A2 may participate in the process of disulfidptosis after SCI through the cytokine-cytokine receptor interaction pathway, MAPK signaling pathway, steroid biosynthesis, glycosaminoglycan degradation, and natural killer cell-mediated cytotoxicity pathways." Thank you for your valuable suggestions.

Best regards.

14. Line 444-447, they are also making claims that are not supported by the data of the paper. The authors should not use the word “confirm” based only on bioinformatic analysis of transcriptomic data.

Dear Reviewer,

Thank you for pointing out the shortcomings of the paper. We have revised the relevant statements based on your feedback. The modified text is as follows: "Strengths of this study include the following aspects. Firstly, we are the first observation in rat, mouse, and human peripheral blood leukocyte samples that disulfidptosis may be involved in the pathophysiological processes following spinal cord injury." Thank you for your valuable suggestions.

Best regards.

15. Line 453-454, the data of this paper does not provide sufficient evidence for this claim.

Dear Reviewer,

Thank you for pointing out the shortcomings of the paper. We have made appropriate modifications in the manuscript based on your feedback. The revised text reads: "In summary, the pathophysiological process following spinal cord injury probably involves disulfidptosis, and a total of three hSCI-DRGs (CAPZB, SCL3A2, and TLN1) have been filtered. Furthermore, all hSCI-DRGs may be implicated in post-SCI pathway regulation, immune infiltration, gene-transcription factor-drug regulatory network. These findings provide novel insights into understanding the pathological mechanisms underlying disulfidptosis in SCI and have implications for its diagnosis, treatment, and future research endeavors."

Best regards.

Answer questions to Reviewer 2

Reviewer 2： I believe this manuscript meets the publication requirements of PLOS ONE. Therefore, I recommend its acceptance pending minor revisions. Below are my comments for the authors。

Dear Reviewer,

Thank you very much for your very unique and precise suggestions on the manuscript. We have diligently improved the manuscript based on your advice. We also hope that the manuscript meets the publication requirements of PLOS ONE. Once again, thank you for your valuable input.

Best regards.

1. The figure legends should be reworded. I suggest assigning clear titles to each figure and describing each panel separately.

Dear Reviewer,

Thank you for your suggestions. We have rewritten all figure captions in more detail.

Best regards.

2. Once a term is abbreviated, use only the abbreviation in subsequent mentions (examples: lines 47–56, 66–100, 38–79).

Dear Reviewer,

Thank you for your reminder. We have carefully checked the abbreviations in the manuscript based on your suggestion and removed any unnecessary ones.

Best regards.

3. I recommend adding a sentence at the end of the last paragraph of the introduction to summarize the major takeaway of the study.

Dear Reviewer,

Thank you for your suggestions. At the end of the introduction, we have rephrased the language and briefly summarized the findings of the article. The added content reads:

"To gain a comprehensive understanding of the gene regulatory mechanisms associated with disulfidptosis in SCI patients, we employed bioinformatics technology and experimental verification to identify three genes (CAPZB, SLC3A2, and TLN1) that are imp

---

## [Decision Letter · Decision Letter 1]

8 Jan 2025

Bioinformatics analysis of genes associated with disulfidptosis in spinal cord injury

PONE-D-24-34139R1

Dear Dr. Tian,

We’re pleased to inform you that your manuscript has been judged scientifically suitable for publication and will be formally accepted for publication once it meets all outstanding technical requirements.

Kind regards,

Zhiwen Luo

Academic Editor

PLOS ONE

Additional Editor Comments (optional):

Reviewers' comments:

Reviewer's Responses to Questions

**Comments to the Author**

1. If the authors have adequately addressed your comments raised in a previous round of review and you feel that this manuscript is now acceptable for publication, you may indicate that here to bypass the “Comments to the Author” section, enter your conflict of interest statement in the “Confidential to Editor” section, and submit your "Accept" recommendation.

Reviewer #2: All comments have been addressed

2. Is the manuscript technically sound, and do the data support the conclusions?

Reviewer #2: Yes

3. Has the statistical analysis been performed appropriately and rigorously? 

Reviewer #2: Yes

4. Have the authors made all data underlying the findings in their manuscript fully available?

Reviewer #2: Yes

5. Is the manuscript presented in an intelligible fashion and written in standard English?

Reviewer #2: Yes

6. Review Comments to the Author

Reviewer #2: The authors have addressed my concerns, therefore, I recommend the publication of this manuscript in PLOS ONE, provided that Reviewer 1's concerns are also addressed satisfactorily.

7. PLOS authors have the option to publish the peer review history of their article (what does this mean?). If published, this will include your full peer review and any attached files.

Reviewer #2: No

---

## [Editor Report · Acceptance letter]

14 Jan 2025

PONE-D-24-34139R1 

PLOS ONE

Dear Dr. Tian, 

I'm pleased to inform you that your manuscript has been deemed suitable for publication in PLOS ONE. Congratulations! Your manuscript is now being handed over to our production team.

Kind regards, 

on behalf of

Dr. Zhiwen Luo 

Academic Editor

PLOS ONE